# Mapping of Quantitative Trait Loci Controlling Egg-Quality and -Production Traits in Japanese Quail (*Coturnix japonica*) Using Restriction-Site Associated DNA Sequencing

**DOI:** 10.3390/genes12050735

**Published:** 2021-05-13

**Authors:** Mohammad Ibrahim Haqani, Shigeru Nomura, Michiharu Nakano, Tatsuhiko Goto, Atsushi J. Nagano, Atsushi Takenouchi, Yoshiaki Nakamura, Akira Ishikawa, Masaoki Tsudzuki

**Affiliations:** 1Graduate School of Integrated Sciences for Life, Hiroshima University, Higashi-Hiroshima, Hiroshima 739-8525, Japan; m190772@hiroshima-u.ac.jp (S.N.); minaka@hiroshima-u.ac.jp (M.N.); takeatsu@hiroshima-u.ac.jp (A.T.); ynsu@hiroshima-u.ac.jp (Y.N.); 2Research Center for Global Agromedicine, Obihiro University of Agriculture and Veterinary Medicine, Obihiro, Hokkaido 080-8555, Japan; tats.goto@obihiro.ac.jp; 3Japanese Avian Bioresource Project Research Center, Hiroshima University, Higashi-Hiroshima, Hiroshima 739-8525, Japan; 4Faculty of Agriculture, Ryukoku University, Otsu, Shiga 520-2194, Japan; anagano1234@gmail.com; 5Graduate School of Bioagricultural Sciences, Nagoya University, Nagoya, Aichi 464-8601, Japan

**Keywords:** egg-related traits, QTL, RAD-seq, SNP marker, Japanese quail

## Abstract

This research was conducted to identify quantitative trait loci (QTL) associated with egg-related traits by constructing a genetic linkage map based on single nucleotide polymorphism (SNP) markers using restriction-site associated DNA sequencing (RAD-seq) in Japanese quail. A total of 138 F_2_ females were produced by full-sib mating of F_1_ birds derived from an intercross between a male of the large-sized strain with three females of the normal-sized strain. Eggs were investigated at two different stages: the beginning stage of egg-laying and at 12 weeks of age (second stage). Five eggs were analyzed for egg weight, lengths of the long and short axes, egg shell strength and weight, yolk weight and diameter, albumen weight, egg equator thickness, and yolk color (L*, a*, and b* values) at each stage. Moreover, the age at first egg, the cumulative number of eggs laid, and egg production rate were recorded. RAD-seq developed 118 SNP markers and mapped them to 13 linkage groups using the Map Manager QTX b20 software. Markers were spanned on 776.1 cM with an average spacing of 7.4 cM. Nine QTL were identified on chromosomes 2, 4, 6, 10, 12, and Z using the simple interval mapping method in the R/qtl package. The QTL detected affected 10 egg traits of egg weight, lengths of the long and short axes of egg, egg shell strength, yolk diameter and weight, albumen weight, and egg shell weight at the beginning stage, yellowness of the yolk color at the second stage, and age at first egg. This is the first report to perform a quail QTL analysis of egg-related traits using RAD-seq. These results highlight the effectiveness of RAD-seq associated with targeted QTL and the application of marker-assisted selection in the poultry industry, particularly in the Japanese quail.

## 1. Introduction

Tremendous progress has been made in poultry egg production and breeding programs over the past decades, leading to the fast-growing livestock industry. In the future, increased advances are required to enhance egg-quality traits in particular. Egg-related traits are complex traits controlled by multiple quantitative trait loci (QTL) and are influenced by genetic and environmental factors and their interactions [1]. QTL analysis can increase the selection response in breeding programs by investigating genotype and phenotype relationships using marker-assisted selection (MAS) [2,3]. Several DNA markers, such as single nucleotide polymorphisms (SNPs) [4], restriction fragment length polymorphisms [5], amplified fragment length polymorphisms (AFLPs) [6], and simple sequence length polymorphisms (or microsatellite) [7], have been used to identify the chromosomal positions of loci. Among these markers, SNPs play a crucial role in genetics, biomedicine, ecology, and evolutionary studies [8,9,10,11]. The other types of DNA markers are associated with a limited number of available markers, high cost of genotyping, and relatively large distance between marker and QTL [12]. In contrast, SNP markers show a stably inherited mechanism, provide more potential markers near or in any locus of interest, are located in coding regions, directly affect protein function, and are more suitable for throughput genetic analysis [13]. It has been difficult to obtain increased markers in the last few years owing to labor and cost-intensive approaches [14]. Next-generation sequencing technologies with recent advances in genotyping by sequencing (GBS) methods and related bioinformatics-computing resources facilitate the large-scale, rapid, and cost-effective discovery of SNPs [15,16,17,18,19]. Restriction-site associated DNA sequencing (RAD-seq) is a GBS method that can simultaneously identify, verify, and score thousands of SNPs, reduce complexity across genomes, deliver high-resolution population genomic data, and be convenient for non-model species at a reasonable cost [20,21,22,23]. 

The Japanese quail (*Coturnix japonica*), similar to chicken, belongs to the order Galliformes and the family *phasianidae*, is a model bird used for production [24] and experimental purposes [25,26]. The Japanese quail as an oviparous animal has a small body size, shows a rapid turnover of generations, has inexpensive rearing requirements, is adaptable to a wide range of husbandry conditions, is easy to handle, has fast growth performance, shows resistance to diseases, and is a more efficient converter of feed to eggs than chickens and can lay approximately 300 eggs per year [27,28]. 

Quail eggs are a suitable source of animal protein and fulfill the nutritional requirements of the growing world population. These eggs contain all essential amino acids for humans and provide a sufficient amount of several vitamins and minerals. External and internal egg-related traits are the most important egg traits that influence egg quality, hatching performance, body weight of newly hatched chicks, propagation of flocks, and breeding economy of the poultry industry [29]. These factors highlight the importance of egg quality in modern poultry breeding conditions. In the current study, we used large-sized (LS) and normal-sized (NS) Japanese quail strains as parents to develop an F_2_ resource population. The LS Japanese quail has a large body weight and is used for meat production purposes. NS Japanese quails have a normal body weight and are used for egg production [30]. 

Compared with chickens, few studies have performed QTL analysis on the Japanese quail, and the genome of this species has been poorly explored [31]. The first studies reported on the genetic and QTL analyses of the Japanese quail involved the identification of panels of markers [32,33], construction of genetic maps using AFLP [34] and microsatellite [35], complete sequencing of the mitochondrial genome [36], detection of QTL for growth-performance and meat-quality traits [37], determination of egg-laying curve [38], and analyses of carcass traits, internal organs [39], and fearfulness-related traits [6,40]. The construction of genetic maps and QTL analysis in poultry have mostly focused on chickens rather than Japanese quails. Recently, existing reports on the genetic map of the Japanese quail were integrated and aligned to the assembled chicken sequence data [41]. As the Japanese quail shows close phylogenetic relatedness to chickens with similar length of the genome (1.2 × 10^9^ base pairs) and chromosome number (2n = 78) and homology of chromosome morphology, a high rate of synteny conservation is expected between the two species [42]. According to the chicken QTL database (QTLdb) [43], 11,818 QTL responsible for 420 different traits have been detected in 318 studies [44,45,46,47,48,49,50]. On the contrary, few studies have reported the QTL responsible for egg-related traits in the Japanese quail. Knaga et al. [31] revealed four QTL on chromosome 1 that affect egg number, egg production rate, egg weight, and egg shell weight. Recoquillay et al. [51] found four QTL for the mean egg weight located on chromosomes 1, 3, and 18, two QTL for the number of eggs laid positioned on chromosomes 3 and 18, and two QTL for the age at first egg placed on chromosomes 3 and 19 in a study of medium-density genetic map and QTL analyses of the Japanese quail, in which egg weight as an egg-quality trait and two egg production traits were investigated. Minvielle et al. [52] identified QTL for egg weight, egg number, and age at first egg on chromosome 6 and eggshell weight on chromosomes 1, 5, and 20 using microsatellite markers. 

Next-generation sequencing [53] and the construction of high-resolution linkage maps based on SNP markers have been performed in the Japanese quail [6,51]. To the best of our knowledge, there are no published reports on QTL analysis of egg-related traits using RAD-seq in the Japanese quail. However, genetic analysis studies using the RAD-seq method in chickens [54,55], aquaculture [56,57,58], and mammals [59,60] have been reported previously. Hence, the aim of this study was to detect QTL affecting egg-quality and -production traits by constructing a genetic linkage map based on SNP markers using RAD-seq in the Japanese quail. 

## 2. Materials and Methods

### 2.1. Experimental Birds

LS and NS quail strains were reared at the Research Farm of Hiroshima University, Japan. F_1_ birds were obtained from an intercross of one LS male with three NS females. Subsequently, 138 F_2_ females were produced from an intercross between three F_1_ males and nine F_1_ females. In addition to the above three parents and nine F_1_ females, the phenotypic values of 97 parents and 16 F_1_ females were used for comparison. Chick management and food supply were in accordance with a previous study [61]. Newly hatched chicks were leg-banded and weighed before being moved to heated brooders, where they were reared until 4 weeks of age. Thereafter, they were housed in individual cages (depth: 15 cm; width: 18 cm; height: 18 cm). Chicks were fed a standard chick diet (22% crude protein (CP); 2900 kcal metabolizable energy (ME) kg^−1^) ad libitum for ages 0 to 4 weeks, followed by a grower diet (17% CP and 2850 kcal ME kg^−1^) from 4 to 16 weeks of age. The birds were maintained under a 24-h lighting photoperiod for 4 weeks, followed by a 14 h:10 h light:dark cycle. They were reared according to the protocol described in the Guidelines for Proper Conduct of Animal Experiments [62].

### 2.2. Traits

A total of 263 female quails were used to assess egg-related traits. Blood samples from all birds were collected using the method described by Kabir et al. [63]. External and internal egg-related traits were measured at two different egg production stages, at the beginning of the egg production period (first stage) and 12 weeks of age (second stage). The two stages are indicated with the subscript letters _1_ and _2_ in the abbreviations of traits. Five eggs from each quail hen were evaluated at each stage. In general, a total number of 2630 eggs were measured from the parental, F_1_, and F_2_ generations. A total of 1000, 250 and 1380 eggs were evaluated at the parental, F_1_, and F_2_ generations, respectively. External and internal egg-related traits, including egg weight (EW), egg long axis (ELA), egg short axis (ESA), egg shell strength (ESS), egg shell weight (ESW), egg equator thickness (EET), yolk weight (YW), yolk diameter (YD), yolk color, lightness (L* value) (YC-L*), redness (a* value) (YC-a*), and yellowness (b* value) (YC-b*), and albumen weight (AW), as well as the age at first egg (AFE), total number of laid eggs from maturation up to 16 weeks of age (TLE), and egg production rate (EPR) were used for evaluation in this study. 

EW, ESW, YW, and AW were measured using a digital balance GX-2000 (A & D Company Ltd., Tokyo, Japan). The lengths of ELA and ESA, and YD were measured using a digital micrometer ABS Digi-Kanon (Nakamura Mfg. Co. Ltd., Tokyo, Japan). For the ESS, an egg shell strength meter (FHK Fujihira Industry Co. Ltd., Tokyo, Japan) was used. EST was measured using an eggshell thickness micrometer (Mitutoyo Industry Co. Ltd., Tokyo, Japan). Yolk color results were obtained using the international colorimetric system of CIELAB. The calibration was based on a black standard with L* = 0 and a white standard with L* = 100. The balanced CIELAB system was determined by three mutually perpendicular axes L*, a*, and b* defined by lightness, redness, and yellowness of yolk color, respectively. The coordinates a* and b* represent the regions of the spectrum with wavelengths corresponding to colors from green (−a) to red (+a) and from blue (−b) to yellow (+b), respectively. The scale of lightness L* represents an interval from 0 (black) to 100 (white). Thus, the complementary color system was based on the differences in the three elementary color pairs: red/green, yellow/blue, and black/white [64,65]. A Chroma meter CR-300 was used to measure the YC-L*, a*, and b* values (Konica Minolta Holdings Inc., Tokyo, Japan).

### 2.3. RAD Library Preparation and Sequencing

Collected blood samples from all birds were used for the extraction of genomic DNA using the phenol-chloroform method and Qiagen kit (DNeasy Blood & Tissue Kits—QIAGEN, Venlo, The Netherlands), according to the manufacturer’s protocol. The quality of extracted DNA in each sample was measured using a Qubit 3.0 assay fluorometer (Thermo Fisher Scientific Inc. Waltham, MA, USA), and the DNA was finally adjusted to a concentration of 20 ng/µL for RAD-seq. RAD-seq was performed for the 4 parents and 12 F_1_ and 138 F_2_ individuals. The RAD library construction procedure followed the methodology previously described by Sakaguchi et al. [66]. The RAD library was sequenced using HiSeq 2500 (Illumina, San Diego, CA, USA) in a 50-bp single-end adapter using *EcoRI* and *BglII*. The RAD-seq read data were consigned in the DDBJ Sequence Read Archive (accession no. DRA011153). The RAD-seq reads were trimmed using the Trim_Galore program (http://www.bioinformatics.babraham.ac.uk/projects/trim_galore/, accessed on 16 March 2020), and the trimmed reads were mapped onto the Japanese quail genome (GCA_001577835.1 *Coturnix japonica* 2.—NCBI) using Bowtie2 [67]. The resulting binary sequence alignment/map format (BAM) files processed with SAMtools [68] were used for variant detection. Variant detection was initially performed for F_1_ strains. The BAM files of F_1_ strains were processed using SAMtools mpileup and varscan2 mpileup2cns [69] with default parameters but changed to min-coverage 5. The variant call format (vcf) files were merged with bcftools [70], and the merged vcf file was further screened using vcftools [71] with the following parameters: minDP 5, min-meanDP 5, maxDP 100, min-alleles 2, and max-alleles 2. The screened sites, which were heterozygous for all F_1_ strains, were summarized in the position list. With the position list, polymorphisms of all samples, including strains of P_1_, P_2_, F_1_, and F_2_, were called using samtools mpileup and varscan with the above parameters and merged using bcftools. After the polymorphism detection steps, only the GT fields were exported using vcftools and used for further analysis. 

The RAD-seq reads that showed different alleles between the two parental strains and genotyped in more than 90% of the F_2_ resource population were selected for mapping. The Map Manager QTX b20 application was used to construct the linkage map [72]. The Kosambi map function was used to calculate the genetic distance with linkage criterion *p* = 0.001 [73].

### 2.4. QTL Analysis

The scanone function of R/qtl [74] was used for QTL analysis using the simple interval mapping method [75]. The genome-wide significance (1% and 5%) and suggestive at (10%) threshold levels were determined by 1000 permutation tests [76]. The significant thresholds for the Z chromosome were estimated using the method described by Broman et al. [77]. The confidence interval from the logarithm of odds (LOD) drop-off method was calculated as 1.8 in this study [78]. The statistical analysis for the calculation of confidence interval, percentage of phenotypic variance, and additive and dominant effects of QTL was conducted using R/qtl based on a previous study [76].

### 2.5. Statistical Analysis

Egg-related data were adjusted for the effects of birth date and dams using the least squares method in R-Project for Statistical Analysis v. 3.6.1. [79]. Egg-related trait comparisons among parental, F_1_, and F_2_ generations were performed using one-way analysis of variance (ANOVA), followed by Tukey’s HSD test with JMP v. 11.0.2 (SAS Institute Inc., Tokyo, Japan).

## 3. Results

### 3.1. Phenotypic Values

The estimated means and standard errors for egg-related traits are shown in Table 1. The LS birds presented the highest values for EW_1_, ELA_1_, ESA_1_, AW_1_, YW_1_, ESW_1_, EW_2_, ESA_2_, AW_2_, and ESW_2_ traits. Significant differences were observed for the YC-a*_1_ and ESA_2_ traits between each generation. The highest values were detected for YC-a*_1_ in NS, both stages of YC-L* in F_1_, and YC-b*_1_, YD_2_, and AFE in the F_2_ generation. On the contrary, the lowest values of ESA_2_, YD_2_, and YW_2_ for NS and YC-a* in both stages and ESS_2_ for F_2_ generation were observed. No significant difference was observed between the parents and the F_1_ generation for ESA_2_, TLE, and EPR. In addition, no significant difference was observed for EW_1_, ELA_1_, ESA_1_, and YW_1_ between the NS and both filial generations. 

### 3.2. RAD Sequencing and SNP Markers

The Illumina HiSeq 2500 yielded 123,700,031 RAD-seq reads for F_2_ birds. After removing the uninformative markers, a total of 25,631 SNP markers were identified. Four hundred twenty-five SNPs were excluded that did not fit the Chi-squared goodness-of-fit test (*p* < 0.05). Out of the remaining 25,206 SNPs, 9843 markers were discarded that genotyped at least 90% of all F_2_ individuals. After stringent filtration, 14,659 SNPs with a low level of heterozygosity in the F_1_ parents were removed. After quality-based filtering, 571 SNPs were discarded from further analyses to reduce missing and wrongly called SNPs. A total of 133 high-fidelity SNP with fixed genotypes in both parents were obtained after excluding those with a deviation for the Mendelian segregation pattern. The QTX Map manager detected 15 unlinked markers that were excluded. Finally, 118 SNP markers were found to be informative between the parental strains.

### 3.3. Linkage Map

A linkage map was successfully drawn using the selected markers in Figure 1. The summary statistics of the constructed genetic map shown in Table 2. Linkage maps contained 118 SNP markers arranged in 13 linkage groups. The chromosome Z in this study was divided into two linkage groups. Each linkage group contained 2 to 32 SNPs. The length of Linkage groups ranged from 0 to 205.3 cM. SNPs covered 776.1 cM of total genetic length with an average spacing of 7.4 cM. 

### 3.4. QTL Detection

Nine main-effect QTL were detected on chromosomes 2, 4, 6, 10, 12, and Z (Table 3). Genome-wide significant and suggestive levels for simple interval mapping calculated were 2.963–4.562 in LOD score for the detected QTL. Genome-wide LOD plots for detected QTL are shown in Figure 2, Figure 3, Figure 4 and Figure 5. The plot of a phenotype against the genotypes at a marker for the identified QTL was drawn in Appendix A. Two genome-wide significant and suggestive QTL were found on chromosome 2 for AFE and YC-b*_2_, respectively. QTL for AFE was located at 108.0 cM with 10.4% of phenotypic variance, and YC-b*_2_ QTL positioned 123.5 cM with a phenotypic variance of 9.9%. A single genome-wide suggestive QTL was detected for ELA_1_ located at 13.3 cM on chromosome 4. The detected QTL on chromosome 6 were located at 54.0 and 69.0 cM, which were responsible for YW_1_ and ESS_1_, respectively. Discovered QTL on chromosome 10 affected EW_1_, ESA_1_, AW_1_, and YW_1_ traits that ranged in the same flanking markers (0.0–26.0 cM). A genome-wide suggestive QTL was found for ESW_1_ located on chromosome 12 between the confidence interval of 4.0–42.0 cM with 10.0% of phenotypic variance and positive additive (0.063) and dominant (0.041) effects. The Z chromosome presented two QTL for ESA_1_ and YD_1_ positioned at 49.4 and 52.0 cM, respectively. Detected QTL are shown in the genetic map in Figure 6. 

## 4. Discussion

The present study revealed nine QTL for 10 egg-related traits (EW_1_, ELA_1_, ESA_1_, ESS_1_, YD_1_, AW_1_, YW_1_, ESW_1_, YC-b*_2_, and AFE) in 138 F_2_ female Japanese quails. To the best of our knowledge, four studies have conducted QTL analysis on egg-quality and -production traits in the Japanese quail. In these studies, fewer egg-related traits and markers were used for QTL analysis. As the Japanese quail is a model bird in Galliformes, the data obtained in quail can be applied to other bird species, especially in chickens. In addition, the Japanese quail shows close phylogenetic relatedness to chickens with similar length of genome, chromosome number, and morphology of chromosomes. Therefore, a high rate of contiguity, assembly statistics, gene content, chromosomal organization, and synteny conservation exists between the two species [42,80]. Knaga et al. [31] reported QTL for the egg number and egg production rate at positions 36 to 42 cM on chromosome 1 using 30 microsatellite markers. Minvielle et al. [52] reported QTL for egg number and age at first egg positioned at 32 and 34 cM, respectively, on chromosome 6. Recoquillay et al. [51] reported QTL for egg number on chromosomes 3 (225 cM) and 18 (3 cM) and QTL for age at first egg on chromosomes 3 (302 cM) and 19 (4 cM). The results of the current study detected a QTL for AFE located on chromosome 2. However, we did not reveal any QTL for EPR and TLE in the present study. The discrepancy in the number of laid eggs between the results of the present study and those of previous studies may be related to the different durations of the egg collection period. The results of this study are in line with those of Goraga et al. [81], who found QTL for AFE on chromosome 2 in chickens. 

The present study found a genome-wide significant QTL for EW at the first stage of egg-laying on chromosome 10 positioned at 21.0 cM. A QTL responsible for EW was reported on chromosome 1 at a position of 4 cM for the Japanese quail [31]. A previous study identified four QTL for the EW trait on chromosomes 1, 3, 18, and 18 positioned at 193, 156, 61, and 62 cM, respectively [51]. At position 0 cM on chromosome 6, a QTL mapped for EW was reported in Japanese quail eggs [52]. The differences in the positions and chromosome numbers of detected QTL for EW may be related to the different periods of weight determination and suggest that other genes exert an effect during the laying period rather than at a later age [31]. In chickens, QTL for the first egg weight were determined on chromosomes 4 and 8 at 213 and 40 cM, respectively [47]. In addition, QTL underlying EW at later ages have been reported on chromosomes 8 and 10 in chickens [47,82]. 

A genome-wide suggestive QTL was found for ESW at the first stage of egg-laying on chromosome 12 in the current study. In turn, studies of ESW in the Japanese quail have reported QTL on chromosome 1 [31] and chromosomes 1, 5, and 20 [52]. A similar result was found in the case of chickens in which QTL associated with ESW was mapped on chromosome 12 with a confidence interval of 0–71 cM [83]. The discrepancy between the findings of QTL for ESW in the Japanese quail may be due to different times during the production period for determining ESW or the lack of a single preferred way to measure the shell weight. Therefore, it is important to identify QTL that differ over time for the traits of interest. 

To date, no QTL have been detected for AW and YW in Japanese quail eggs. This study is the first to report QTL for AW and YW. However, QTL detection for AW and YW has been reported in chickens to be located on chromosomes 2, 3, 4, 7, and 8 for AW and 1–6, 8, 9, 11, 13, 15, 17, 22, 23, 26, 28, and Z chromosomes for YW [12,46,50,82,84,85]. No QTL were reported for AW and YW on chromosome 10 in chickens. In the present study, the detected QTL for YW on chromosome 6 did not overlap with the previous study in chickens [82]. This might be due to the differences in crosses designed for both studies or ages of the two species used, as genetic control for YW is age-dependent. The AW and YW traits are influenced by the age of quails. AW and YW increase with the age of the females. For eggs of the same size, old quails produce larger AW and YW than do young quails. However, in chickens, the genetic correlations among AW and YW at different age points are relatively high [46,85].

To date, there have been no studies detecting QTL for ELA, ESA, ESS, YD, and YC in Japanese quail eggs. This is the first study to detect QTL for ELA_1_, ESA_1_, ESS_1_, YD_1_, and YC-b*_2_ traits in the Japanese quail. Nevertheless, there are available QTL for these traits in chickens. Sasaki et al. [86] found a QTL underlying ELA on the same chromosome as that in the current study. The presence of QTL for ESA at the early stage of egg production was reported on chromosomes 10 and Z positioned at 4 and 85 cM, respectively [47]. The positions of the two QTL for ESA are not in line with the positions of QTL on chromosomes 10 and Z for ESA_1_ in the current investigation. Similar to the present study, QTL for ESS have been reported on chromosome 6 in chickens [82]. However, Goto et al. [47] found a QTL for ESS on the Z chromosome at the beginning stage of egg production, positioned at 51 cM. In the present study, a QTL for YD_1_ was mapped on the Z chromosome. However, no QTL have been reported for yolk index on this chromosome in chickens. Detected QTL for the yolk index in chickens were located on chromosomes 2, 8, and 17 [46,87]. The presence of novel QTL and mapping information is very important for understanding the genetic basis of egg-related traits. 

An important biophysical parameter of egg yolk is its color, which plays a very important role in the perception of food and is also a key aspect of food quality. In the current study, one QTL for the yellowness of yolk color at the second stage of egg laying was detected on chromosome 2. QTL for yolk color were also identified on chromosomes 1–4, 8, 9, and 27 in chickens [46,50,88,89]. There is a discrepancy between the findings of this study and those of chickens. It is suggested that the different methods used for the measurement of YC in the current study and the literature on chickens may account for the inconsistent results. 

The findings of this investigation suggest that egg-related traits are regulated by many QTL at different growth stages associated with the importance of egg quality and production in breeding programs for poultry industries. These findings may help to develop the Japanese quail QTL database, which can be applied to other Galliformes species, especially chickens. Detected QTL affecting egg-related traits provides necessary information for further molecular studies to improve quantitative traits. It would be desirable to perform a deeper analysis of detected loci using an expanded dataset and sequence information to assess the potential of the found QTL for MAS. Japanese quail can be a valuable resource for the poultry industry sector, and the results of the current study may enable poultry breeders and industries to develop good breeding strategies based on future MAS studies. In conclusion, we determined nine main-effect QTL that affect egg-related traits using LS and NS Japanese quail strains. This is the first study to report quail QTL information determined using RAD-seq method, including the detection of many trait-regulating QTL that have never been reported before. Thus, these findings will help us understand the genetic basis of egg-related traits and assist breeders in implementing effective selection programs based on MAS.

## Figures and Tables

**Figure 1 genes-12-00735-f001:**
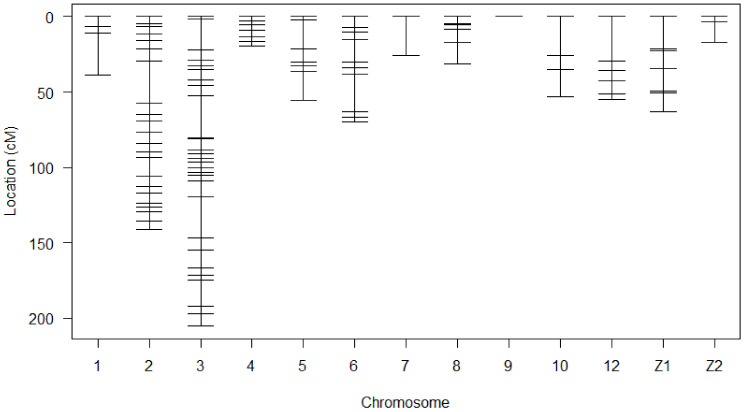
Genetic map of SNP markers distributed on 13 linkage groups. The abscissa shows the chromosome number. The ordinate presents the marker position based on cM.

**Figure 2 genes-12-00735-f002:**
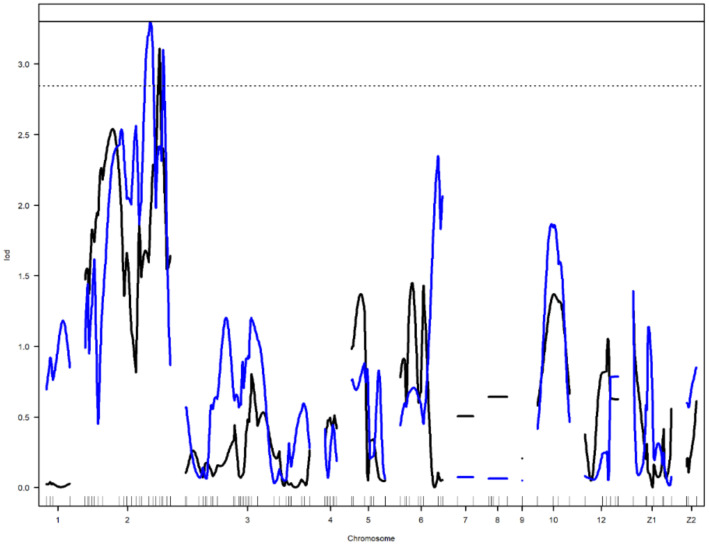
Genome-wide LOD plot of detected QTL for age at first egg (blue) and yolk color-yellowness_2_ (black) on chromosome 2. The horizontal lines show the genome-wide significance at 5% (straight line) and suggestive threshold at 10% (dotted line) levels.

**Figure 3 genes-12-00735-f003:**
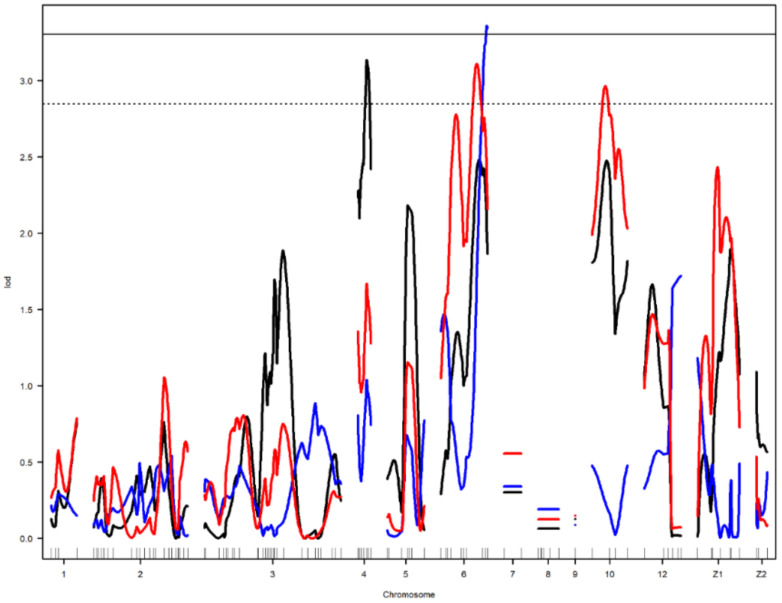
Genome-wide LOD plot of detected QTL for egg long axis_1_ (black) on chromosome 4, yolk weight_1_ (red) on chromosomes 6 and 10, and egg shell strength_1_ (blue) on chromosome 6. The horizontal lines show the genome-wide significance at 5% (straight line) and suggestive threshold at 10% (dotted line) levels.

**Figure 4 genes-12-00735-f004:**
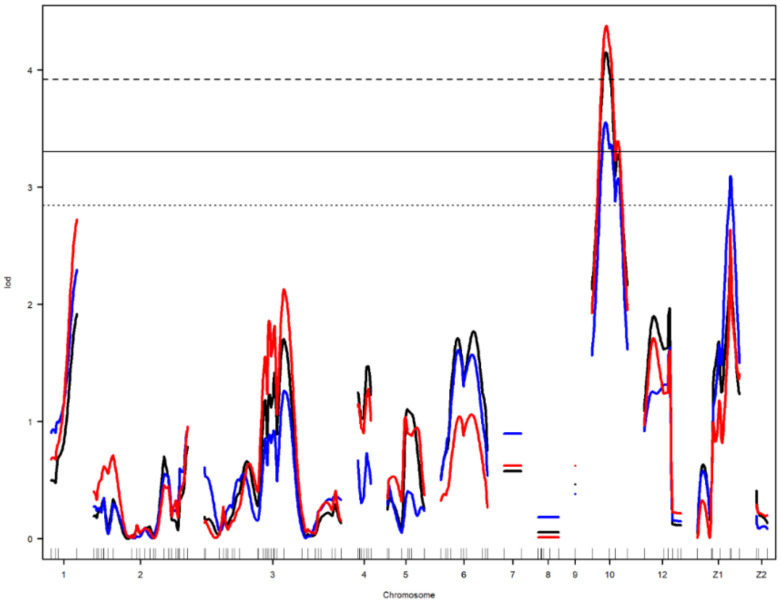
Genome-wide LOD plot of detected QTL for egg weight_1_ (black) and albumen weight_1_ (red) on chromosome 10, and egg short axis_1_ (blue) on chromosomes 10 and Z1. The horizontal lines show the genome-wide significance threshold at 1% (dashed line) and 5% (straight line) and suggestive threshold at 10% (dotted line) levels.

**Figure 5 genes-12-00735-f005:**
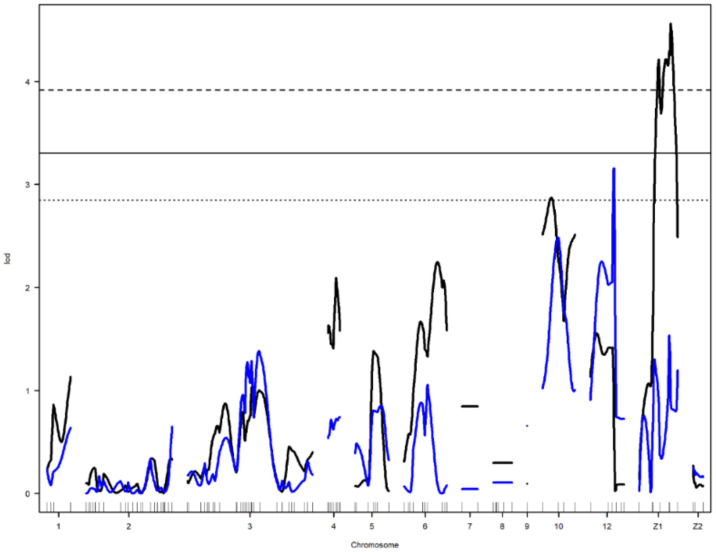
Genome-wide LOD plot of detected QTL for egg shell weight_1_ (blue) on chromosome 12 and yolk diameter_1_ (black) on chromosome Z1. The horizontal lines show the genome-wide significance threshold at 1% (dashed line) and 5% (straight line) and suggestive threshold at 10% (dotted line) levels.

**Figure 6 genes-12-00735-f006:**
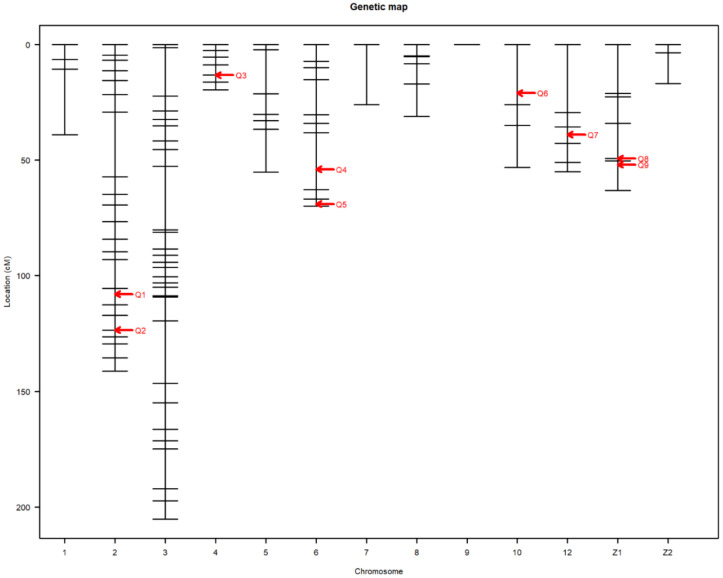
Detected QTL presented in the genetic map. The abscissa shows the chromosome number. The ordinate represents the position of the markers based on cM. Q stands for the QTL number.

**Table 1 genes-12-00735-t001:** Egg-related traits (mean ± standard error) used in this study.

Traits	LS (*n* = 50)	NS (*n* = 50)	F_1_ (*n* = 25)	F_2_ (*n* = 138)
Egg weight_1_ (g)	10.065 ± 0.113 ^a^	8.508 ± 0.107 ^b^	9.001 ± 0.1715 ^b^	8.616 ± 0.0771 ^b^
Egg long axis_1_ (mm)	30.767 ± 0.144 ^a^	29.990 ± 0.185 ^b^	29.803 ± 0.2095 ^b^	29.754 ± 0.1052 ^b^
Egg short axis_1_ (mm)	24.754 ± 0.094 ^a^	23.497 ± 0.097 ^b^	23.746 ± 0.1295 ^b^	23.460 ± 0.0687 ^b^
Egg shell strength_1_ (kg/cm^2^)	1.392 ± 0.037 ^b^	1.519 ± 0.041 ^a^	1.446 ± 0.0319 ^ab^	1.219 ± 0.0160 ^c^
Yolk diameter_1_ (mm)	23.783 ± 0.148 ^a^	23.132 ± 0.206 ^b^	23.107 ± 0.1925 ^b^	23.983 ± 0.1003 ^a^
Albumen weight_1_ (g)	5.482 ± 0.067 ^a^	4.511 ± 0.057 ^c^	4.914 ± 0.0852 ^b^	4.664 ± 0.0430 ^bc^
Yolk weight_1_ (g)	3.082 ± 0.048 ^a^	2.900 ± 0.051 ^b^	2.829 ± 0.0531 ^b^	2.843 ± 0.0291 ^b^
Egg shell weight_1_ (g)	1.166 ± 0.013 ^a^	1.023 ± 0.014 ^bc^	1.081 ± 0.0211 ^b^	0.979 ± 0.0099 ^c^
Egg equator thickness_1_ (mm)	0.283 ± 0.003 ^ab^	0.279 ± 0.003 ^b^	0.293 ± 0.0025 ^a^	0.285 ± 0.0016 ^ab^
Yolk color-lightness_1_	56.234 ± 0.219 ^b^	56.506 ± 0.293 ^b^	58.317 ± 0.3500 ^a^	55.782 ± 0.1431 ^b^
Yolk color-redness_1_	9.068 ± 0.268 ^b^	10.531 ± 0.348 ^a^	5.327 ± 0.5274 ^c^	2.636 ± 0.2340 ^d^
Yolk color-yellowness_1_	36.519 ± 0.341 ^b^	36.252 ± 0.365 ^b^	34.338 ± 0.3670 ^c^	38.252 ± 0.2120 ^a^
Egg weight_2_ (g)	12.152 ± 0.124 ^a^	9.905 ± 0.116 ^c^	10.846 ± 0.168 ^b^	10.059 ± 0.100 ^c^
Egg long axis_2_ (mm)	31.958 ± 0.136 ^ab^	31.442 ± 0.175 ^b^	31.665 ± 0.187 ^ab^	32.045 ± 0.116 ^a^
Egg short axis_2_ (mm)	26.175 ± 0.104 ^a^	24.087 ± 0.095 ^d^	25.181 ± 0.170 ^b^	24.566 ± 0.077 ^c^
Egg shell strength_2_ (kg/cm^2^)	1.479 ± 0.053 ^a^	1.523 ± 0.049 ^a^	1.480 ± 0.040 ^a^	1.209 ± 0.014 ^b^
Yolk diameter_2_ (mm)	25.164 ± 0.126 ^b^	23.650 ± 0.154 ^c^	24.985 ± 0.164 ^b^	26.135 ± 0.109 ^a^
Albumen weight_2_ (g)	6.270 ± 0.076 ^a^	5.126 ± 0.069 ^c^	5.706 ± 0.079 ^b^	5.367 ± 0.056 ^c^
Yolk weight_2_ (g)	3.770 ± 0.048 ^a^	3.262 ± 0.050 ^b^	3.526 ± 0.057 ^a^	3.701 ± 0.038 ^a^
Egg shell weight_2_ (g)	1.367 ± 0.016 ^a^	1.201 ± 0.017 ^bc^	1.255 ± 0.020 ^b^	1.155 ± 0.013 ^c^
Egg equator thickness_2_ (mm)	0.298 ± 0.003 ^ab^	0.284 ± 0.003 ^c^	0.308 ± 0.003 ^a^	0.291 ± 0.002 ^bc^
Yolk color-lightness_2_	58.464 ± 0.258 ^b^	58.838 ± 0.290 ^b^	60.417 ± 0.351 ^a^	58.602 ± 0.162 ^b^
Yolk color-redness_2_	8.897 ± 0.260 ^a^	7.918 ± 0.371 ^ab^	6.238 ± 0.508 ^b^	2.999 ± 0.256 ^c^
Yolk color-yellowness_2_	41.081 ± 0.357 ^a^	41.231 ± 0.346 ^a^	38.221 ± 0.478 ^b^	37.155 ± 0.217 ^b^
Age at first egg	46.120 ± 0.505 ^b^	44.240 ± 0.923 ^bc^	42.280 ± 0.621 ^c^	49.007 ± 0.444 ^a^
Total laid eggs	56.900 ± 1.169 ^a^	57.000 ± 1.430 ^a^	59.240 ± 2.310 ^a^	47.819 ± 0.782 ^b^
Egg production rate	0.862 ± 0.015 ^a^	0.838 ± 0.016 ^a^	0.849 ± 0.032 ^a^	0.759 ± 0.011 ^b^

^a–d^ Means with different superscript letters are significantly different between the traits (Tukey’s HSD test, *p* < 0.05). _1,2_ Subscript letters are first and second egg laying stages.

**Table 2 genes-12-00735-t002:** Summary of the genetic map constructed in this study.

Chromosome No. ^1^	No. of Markers	Genetic Length (cM)	Average Spacing (cM)	Maximum Spacing (cM)
1	4	39.0	13.0	28.3
2	23	141.3	6.4	28.0
3	32	205.3	6.6	27.5
4	7	19.7	3.3	4.5
5	8	55.3	7.9	19.1
6	10	69.9	7.8	24.6
7	2	26.0	26.0	26.0
8	7	31.1	5.2	14.0
9	2	0.0	0.0	0.0
10	4	53.3	17.8	26.0
12	6	55.1	11.0	29.5
Z1	9	63.2	7.9	21.2
Z2	4	16.9	5.6	13.3
overall	118	776.1	7.4	29.5

^1^ Chromosome Z was divided into two linkage groups.

**Table 3 genes-12-00735-t003:** Summary of QTL detected for egg-related traits.

QTL #	Traits	Chr.	Position (cM) ^1^	Flanking Markers (cM) ^2^	LOD ^3^	Interval (cM) ^4^	Var. ^5^ (%)	Add. ^6^ ± SE	Dom. ^7^ ± SE	d/a ^8^	NS/LS ^9^
1	Age at first egg	2	108.0	105.6–112.6	3.295 *	13.0–138.0	10.4	−1.17 ± 0.544	−2.784 ± 0.839	2.387	A/B
2	Yolk color-yellowness_2_	2	123.5	123.5	3.108 ^†^	0.0–141.3	9.9	−1.02 ± 0.27	−0.218 ± 0.369	0.214	A/B
3	Egg long axis_1_	4	13.3	13.3	3.135 ^†^	0.0–19.7	9.9	0.366 ± 0.125	−0.461 ± 0.188	−1.258	B/A
4	Yolk weight_1_	6	54.0	38.2–62.8	3.11 ^†^	3.0–69.9	8.5	−0.14 ± 0.042	0.136 ± 0.069	−1.008	A/B
5	Egg shell strength_1_	6	69.0	66.9–69.9	3.359 *	53.0–69.9	10.6	0.073 ± 0.02	−0.05 ± 0.027	−0.688	A/B
6	Egg weight_1_	10	21.0	0.0–26.0	4.151 **	2.0–51.0	12.9	−0.47 ± 0.108	−0.322 ± 0.163	0.687	A/B
	Egg short axis_1_	10	20.0	0.0–26.0	3.552 *	2.0–52.0	8.0	−0.34 ± 0.099	−0.273 ± 0.156	0.806	A/B
	Albumen weight_1_	10	21.0	0.0–26.0	4.374 **	5.0–48.0	13.6	−0.27 ± 0.06	−0.199 ± 0.092	0.745	A/B
	Yolk weight_1_	10	20.0	0.0–26.0	2.963 ^†^	0.0–53.3	8.1	−0.15 ± 0.041	−0.1 ± 0.063	0.684	A/B
7	Egg shell weight_1_	12	39.0	35.6–42.7	3.157 ^†^	4.0–42.0	10.0	0.063 ± 0.017	0.041 ± 0.026	0.658	B/A
8	Egg short axis_1_	Z1	49.4	49.4	3.093 ^†^	28.0–63.2	6.8	0.194 ± 0.059			B/A
9	Yolk diameter_1_	Z1	52.0	50.3–63.2	4.562 **	24.0–63.0	14.1	0.396 ± 0.091			B/A

_1,2_ Subscript letters are first and second egg laying stages; ^1^ peak position; ^2^ physical position of the marker; ^3^ genome-wide significant and suggestive QTL (** significant at 1% and * 5% level, ^†^ suggestive at 10% level); ^4^ confidence interval at 1.8-LOD drop-off; ^5^ phenotypic variance explained by QTL; ^6^ additive effect of the QTL; ^7^ dominant effect of the QTL; ^8^ the degree of dominance; ^9^ parental allele.

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
