# Peer review of "Mapping of Quantitative Trait Loci Controlling Egg-Quality and -Production Traits in Japanese Quail (Coturnix japonica) Using Restriction-Site Associated DNA Sequencing"

_genes, 2021, doi:10.3390/genes12050735_

Round 1
Reviewer 1 Report
my comments are attached

Reviewer 2 Report
The authors genotyped 138 female birds of a Japanese quail F2 resource population by intercrossing LS and NS strains using the RAD-Seq and constructed a linkage map to identify the QTLs for egg production traits. I think that some revisions are needed in this manuscript.
1. In this study, the authors identified 25,631 SNPs by RAD-Seq, but used only 118 SNPs for linkage mapping. I would like to know how to determine the screening criteria of the SNPs. For example, if the sample call rate is set to >0.75, how much will the constructed maps and the number of significant QTL change?
2. The authors would indicate which parental allele causes the additive genetic effect in Table 3.
3. Would the authors care to discuss how their results will be applied to MAS of commercial Japanese quails?
